# A natural history museum visitor survey of perception, attitude and knowledge (PAK) of microbes and antibiotics

Julia Zichello[1], Preeti Gupta[1], Monique Scott[1,2], Bella Desai[1], Ruth Cohen[1], Lauri Halderman[3], Susan Perkins[4], Ana Porzecanski[5], Paul J. Planet[6], Yael Wyner[7], Martin Blaser[8], Robert Burk[9], Judy Diamond[10], Rod Kennett[11], Jennifer Borland[12], Rob DeSalle[4]*

1 Education Department, American Museum of Natural History, New York, NY, United States of America, 2 Museum Studies Department, Bryn Mawr College, Bryn Mawr, PA, United States of America, 3 Exhibition Department, American Museum of Natural History, New York, NY, United States of America, 4 American Museum of Natural History, Sackler Institute for Comparative Genomics, New York, NY, United States of America, 5 American Museum of Natural History, Center for Biodiversity and Conservation, New York, NY, United States of America, 6 Division of Infectious Diseases, Perelman School of Medicine & Children's Hospital of Philadelphia, University of Pennsylvania, Philadelphia, PA, United States of America, 7 City College of New York, School of Education, New York, NY, United States of America, 8 Department of Medicine and Microbiology, RBHS, Robert Wood Johnson Medical School, Rutgers University, Piscataway, New Jersey, United States of America, 9 Department of Pediatrics, Albert Einstein University, New York, NY, United States of America, 10 University of Nebraska State Museum, Lincoln, NE, United States of America, 11 Questacon, The National Science and Technology Centre of Australia, King Edward Terrace, Parkes, Australian Capital Territory, Australia, 12 Rockman Et Al, Bloomington, IN, United States of America

* desalle@amnh.org

**Data Availability Statement:** The data are available online at https://sepakiosk.com/.

**Funding:** The research in and the writing of this article was supported by the Office of The Director,

## Abstract

A kiosk-based survey at the American Museum of Natural History in New York City in 2016–2018 allowed us to assess public knowledge of antibiotics and public attitudes toward microbes in museum goers. Over 22,000 visitors from 172 countries and territories answered several carefully designed questions about microbes and antibiotics. These visitors also entered age, gender, and country demographic data that allowed for stratification along these demographic and geographic divisions. Because museum goers are likely to be better informed about these and other science-based topics, the results described here can set a potential upper bound for public knowledge on these topics. Surprisingly, the results of our analysis of museum goers' answers about microbes and antibiotics indicate a substantial lack of familiarity with both topics. For example, overall only about 50% of respondents can correctly identify penicillin as an antibiotic and less than 50% of museum visitors view microbes as beneficial. The results described here suggest that we are perhaps off target with our educational efforts in this area and that a major shift in approach toward more basic microbial topics is warranted in our educational efforts.

National Institutes Of Health under Award Number 5R25OD016514-04. The content is solely the responsibility of the authors and does not necessarily represent the official views of the National Institutes of Health. RD Acknowledges the Fulbright Foundation for support. EB is an employee of Rockman et al (now Rockman et al Cooperative Inc.), and receives support in the form of a salary. The company did not play a role in the study design, data collection and analysis, decision to publish, or preparation of the manuscript.

**Competing interests:** The authors declare no competing interests in the study. Support for EB salary does not alter our adherence to PLOS ONE policies on sharing data and materials.

# Introduction

Public perception, attitudes and knowledge (PAK) about the relationship between microbes and human health are a major focus of health professionals and health educators [1–8]. While public interest in microbiology and infectious disease has always been substantial, even more interest has been generated by the expanding research on human microbiomes (communities of microbes living in and on us) in the past decade [9–16]. And more recently with the Covid 19 (SARS Cov2) pandemic there is an important need for public education of viral biology [17–19]. Microbiome studies in particular have shifted the paradigm for how microbes shape human health away from the simplistic view that microbes act solely as pathogens ('germs", and 'bugs') in the human body towards a more sophisticated view that emphasizes ecological and community interactions amongst microbes in the human body. Health professionals and educators must inform the public of the human health impact of this new microbiome research [15,20–24]. Obtaining basic information on how a lay audience perceives and reacts to these topics is necessary for understanding where to direct educational efforts.

While the initial surveys we conducted for this study addressed a broad array of microbe related topics, here we examine specifically, and in detail, two aspects of public knowledge about microbes. The first addresses the public's familiarity with antibiotics and the second focuses on how the public views microbes. The former—public knowledge about antibiotics—has been extensively addressed with copious published surveys (**S1 File**). The latter concerns general attitudes or familiarity toward microbes, a topic which has not been well-surveyed. While there has been some research in the area of PAK about microbes in general [25–33], the ratio in the literature of research surveys of PAK concerning antibiotics (nearly 100 peer reviewed publications) to microbes in general (on the order of ten) is about ten to one. In this study, we examine several hypotheses (**Table 1**) relevant to people's PAK of microbes and antibiotics. Several sub-hypotheses can be generated from the four general ones listed in **Table 1**. Our approach to testing these hypotheses is also outlined in Table 1 and such tests aid us in approaching the objectives of the study. These objectives are twofold; first to assess the museum goer's knowledge of and familiarity with microbes. Second, we wanted to add to the burgeoning literature concerning general knowledge of antibiotics.

# Materials and methods

## Survey design

Survey questions were administered using a kiosk positioned in the museum's Hall of Biodiversity (**S2 File**). Questions were composed by members of the Education/Exhibition

**Table 1. Hypotheses and tests of public knowledge and attitude to antibiotics and microbes.**

| Specific Hypothesis | Test |
|---|---|
| **H1.** There is no impact of native language on the answers given to the survey questions | Comparing results from countries where primary language is English with other countries' answers |
| **H2.** The public perceives microbes as dangerous and pathogenic | Questions 1 and 2 to address this hypothesis |
| **H3.** The public perceives microbes as Having no beneficial qualities | Questions 1 and 2 to address this hypothesis |
| **H4.** The public can identify an antibiotic(s) in a list of medicinal compounds and antibiotics. | Use of Survey 1, Question 2 and Survey 2, Question 3 to address this hypothesis. |
| **H5.** Knowledge of antibiotics does not differ between countries | Use of Survey 1, Question 2 and Survey 2, Question 3 compared across countries |

The hypothesis is given in the left column and the specific test we used in the right column.

Evaluation team, the AMNH-SEPA (Science Education Partnership Award, NIH) team and the AMNH-SEPA scientific advisory committee. Polling question design was guided by preliminary verbal surveys and prior observations of user kiosk experiences at the AMNH. To compensate for language differences of the international audience of the AMNH, we made the prose of the questions and the physical interaction with the kiosks as simple as possible. We also suggest that if a visitor has enough understanding of the English language to begin interacting with the kiosk, they are more than likely capable of understanding the English in the questions.

Data were collected during a two-year period from February 2016 to January 2018) and two separate surveys were conducted (**Table 2**). While the broader goals of the survey included questions about probiotics, hand sanitizer, antibiotics and other microbe related topics here we focus on antibiotics and general knowledge about microbes. The survey questions relevant to these topics are listed in **Table 2** and posted on the SEPA kiosk website ((http://sepakiosk. com/); **S2 File**) where the raw data for the survey are also available. The specific rationale for each question is given in the table below each question. Some of the questions were designed to have correct and incorrect responses, and others were designed to gain insight into the public's attitudes with respect to microbes, meaning there were no right and wrong answers for these latter questions. Answers to questions in the first survey often guided the questions we asked in the second. For example, Question 2 from the first survey (**Table 2**) was designed to assess whether respondents recognized even one of the most basic antibiotics. If the results of the first survey had indicated advanced widespread ability to recognize basic antibiotics, we planned to delve into antibiotic resistance in the second survey. However, Survey 1 results showed poor public recognition of antibiotics. Therefore, the second survey further explored antibiotic recognition. The American Museum of Natural History Institutional Review Board (AMNH-IRB- FWA00006768) examined the proposed work and survey method and waived consent for all participants in the study.

### Assessing language differences

The native languages of participants in the surveys is broad to say the least. To assess the impact of native language on the Survey answers, we examined in detail the answers to two questions from the surveys. We chose answers from respondents from two countries that are predominantly native English-speaking (United States of America [US; n = 3876] and Australia [AU; n = 329]) and two countries that are predominantly non-native English-speaking and non-Western language based (China [CH; n = 326] and Japan [JP; n = 97]) for comparison. These countries were chosen because of their relatively large sample size compared to other countries. The results of this analysis are in **S3 File**.

### Data analysis

Data from the surveys were parsed into separate files for demographic patterns involving age, country of residence, geographic region of residence or gender (**Table 3**). There are two basic ways to assess the frequency of correct and incorrect responses to the objective questions that we explored. The first is a simple tally of the number of times the answer was chosen, regardless of whether several answers were chosen. For instance, for Survey 1 Question 2 a participant who answered aspirin, valium, Tylenol, penicillin and Benadryl would contribute one incorrect count to the total for each of the incorrect compounds (aspirin, valium, Tylenol and Benadryl) and one correct answer to the total for the correct answer (penicillin). These simple tallies can then be used for comparisons. The second way to count correct and incorrect answers is to recognize that for Survey 1 answering only aspirin is incorrect and answering

**Table 2. Questions on Survey 1 and Survey 2 used in the current study.**

| |
|---|
| **Survey 1** |
| **Question 1."Which two words come to mind when you hear the word microbe?"** |
| **1 "Germ"** |
| **2 "Disease"** |
| **3 "Tiny"** |
| **4 "Beneficial"** |
| **5 "Essential"** |
| **6 "Biodiversity"** |
| *This question was asked to guage the initial impressions of microbes.* |
| **Question 2."Which of these is an antibiotic? Select as many as you like!"** |
| **1 "Aspirin"** |
| **2 "Diazepam (e.g. Valium)"** |
| **3 "Acetaminophen (e.g. Tylenol/Paracetamol)"** |
| **4 "Penicillin"** |
| **5 "Antihistamine (e.g. Benadryl)"** |
| *This is an objective question with a clear correct answer–Penicillin. This question was posed to guage the public's knowledge of antibiotics at a very basic level. If we had obtained a sophisticated response to this question we planned to have proceeded in the second survey to ask a question about resistance.* |
| **Survey 2** |
| **Question 1."Which of the following are true statements about microbes: (check all that apply)."** |
| **1 "Microbes are too small for the naked eye to see"** |
| **2 "Microbes only have one cell."** |
| **3 "Microbes are only in dirty places."** |
| **4 "Microbes are essential for life."** |
| **5 "There are many types of microbes."** |
| *This question was asked to follow up on questions from Survey 1 on the public's general impression of microbes.* |
| **Question 2. "For human health, microbes are:"** |
| **1 "Mostly beneficial"** |
| **2 "About half of them beneficial and half of them harmful"** |
| **3 "Mostly harmful"** |
| **4 "Have no impact on human health"** |
| *This question was asked to guage the public's starting point on what they think a microbe is. This question follows up on one from Survey 1.* |
| **Question 3. "Which of these is an antibiotic? Select as many as you like!"** |
| **1 "Aspirin"** |
| **2 "Valium"** |
| **3 "Tylenol/Paracetamol"** |
| **4 "Penicillin"** |
| **5 "Benadryl"** |
| **6 "Neosporin"** |
| **7 "Azithromycin"** |
| *This question was asked to further assess the surprising result from Survey 1, that most respondents misidentified antibiotics.* |

Possible answers are also shown as well as a rationale (in italics) for the question. We present here only the questions on the survey relevant to the current study. Several other questions on the surveys about probiotics and hand sanitizer were also posed, but for clarity we omit them from this table and refer the reader to http://sepakiosk.com/ for a full list of the questions and responses.

**Table 3. Demographics of the kiosk surveys.**

|  | Survey 1 | Survey 2 |
|---|---|---|
| **Kiosk visits** |  |  |
| Total visits | 18103 | 21300 |
| Valid visits | 9893 | 12721 |
| *Continent or region* |  |  |
| AF (Africa) | 379 | 467 |
| AS (Asia) | 795 | 961 |
| EU (Europe) | 2338 | 3115 |
| IND (India) | 199 | 226 |
| LA (Latin America) | 1016 | 1503 |
| ME (Middle East) | 152 | 167 |
| NA (North America) | 4478 | 5711 |
| PAC (Pacific Region) | 472 | 568 |
| *Gender* |  |  |
| Male | 3724 | 4443 |
| Female | 4828 | 6265 |
| ND | 1281 | 2014 |
| *Age Category (years)* |  |  |
| <13 | 2795 | 3005 |
| 13–25 | 2590 | 3348 |
| 25–45 | 2084 | 3100 |
| 45–65 | 1147 | 1622 |
| >65 | 449 | 588 |
| ND | 771 | 1061 |

Counts for Survey 1 and Survey 2 are shown. ND indicates a respondent who did not enter gender or age.

only penicillin is correct and so on for the other compounds. Any answer with multiple choices like aspirin + penicillin is also technically incorrect. Correct and incorrect answers can be tallied in this way too. We call the former way of counting "raw" and the latter way "accurate".

## Ranking countries for correct answers to antibiotic recognition questions

Responses to the antibiotic recognition questions (Survey 1, Question 2; Survey 2, Question 5) were scored as correct or incorrect for countries that had sample sizes greater than 95. Percentage of total number of respondents with correct answers was then calculated and the seventeen countries with N>95 for Survey 1 and the nineteen countries with N>95 for Survey 2 were ranked based on this percentage.

# Results

## Survey data analysis description

The numbers of participants by country are provided on the website (http://sepakiosk.com/) for this project and in **Fig 1**. In addition, subjects were characterized by geographic region of origin (**Table 3**). The data also can be stratified according to gender (Male [M], Female [F] and no answer) and age (age categories <13 years, 13–25 years, 25–45 years, 45–65 years, >65 years, and no answer). Distribution of the respondents' age and sex in the sample are given in Table 2. Most surveys omit children from reporting of results, however for our analysis here

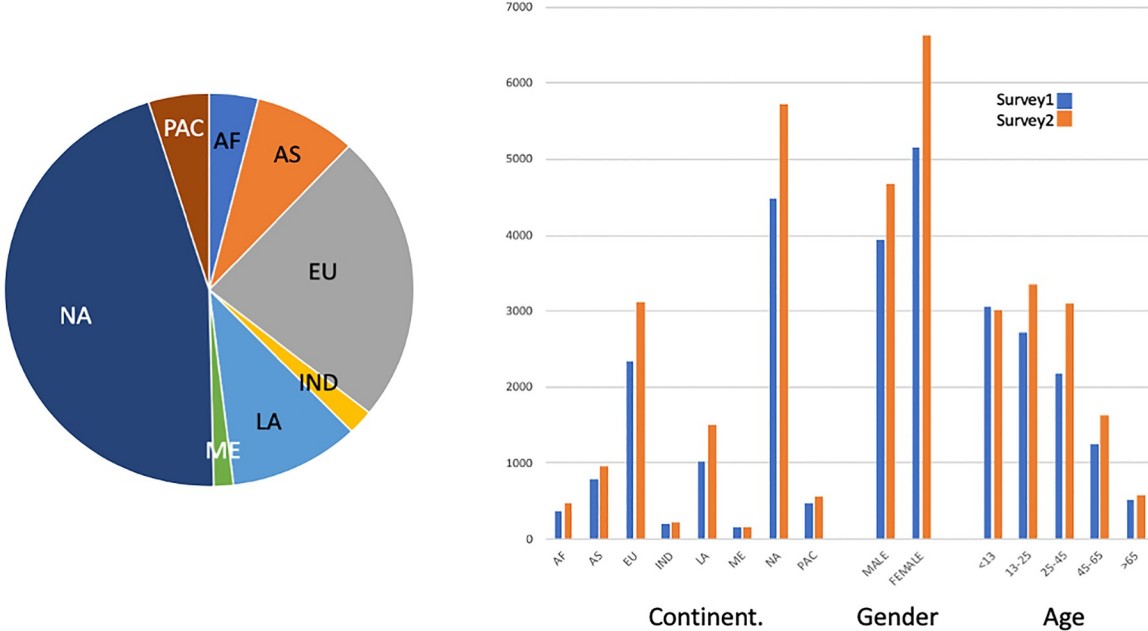

**Fig 1. Demographic characteristics of the survey participants.** The pie chart on the left indicates the geographic distribution of respondents to Survey 1 (Survey 2 showed a similar geographic pattern). The bar chart on the right indicates the numbers of respondents by their geographic residence, gender and age in the two surveys. These characteristics did not vary substantially from Survey 1 to Survey 2 and are similar to the independently collected AMNH visitor patterns over the two years of the survey. The only substantial difference is that international visitors were 5% more likely to visit the kiosk than Americans. Groupings: EU = Europe, NA = North America, ME = Middle East, LA = Latin America, IND = India, AS = Asia, AF = Africa, PAC = Pacific region. AF refers to all countries on the continent of Africa; AS refers to Afghanistan, Azerbaijan, Bhutan, China, Georgia, Indonesia, Japan, Kazakhstan, Kyrgyz Republic, Lao People's Democratic Republic, Malaysia, Mongolia, Myanmar, Nepal, North Korea, South Korea, Taiwan, Tajikistan, Thailand, Turkmenistan, Uzbekistan and Vietnam,. IN refers to Bangladesh, India, Pakistan and Sri Lanka; NA refers to the United States and Canada; EU refers to all European countries; ME refers to Brunei Darussalam, Iran, Iraq, Israel, Jordan, Kuwait, Lebanon, Oman, Saudi Arabia, Syrian Arab Republic, Turkey, United Arab Emirates and Yemen; PAC refers to Australia, Fiji, French Polynesia, New Caledonia, New Zealand, Papua New Guinea, Philippines, Solomon Islands, and Vanuatu; LA refers to Antigua and Barbuda, Argentina, Bolivia, Brazil, Chile, Colombia, Costa Rica, Cuba, Dominica, Dominican Republic, Ecuador, El Salvador, Falkland Islands, Guatemala, Guyana, Haiti, Honduras, Jamaica, Mexico, Nicaragua, Panama, Paraguay, Peru, Saint Kitts and Nevis, Sao Tome and Principe, Suriname, Trinidad and Tobago, Uruguay and Venezuela.

we retained data for respondents <13 years, as the answers for this age did not vary significantly from older respondents. We do, however, recognize the limitations of presenting this age category and so we clearly demarcate this age group in comparisons. For geographic abbreviations used in the figures and the text see the legend to **Fig 1**.

## The effect of language on survey inferences

The results of the analysis of the effects of the use of English on the surveys are presented in **S3 File**. Briefly, the data and analysis show that there are small differences between the respondents from the different countries based on language. However, analysis of language usage also revealed greater variation of answers related to medical compound names, but less with choice of descriptive words (like beneficial or essential). This result is not surprising as the compounds we listed in the survey often times have different brand names in different countries. The disparity in correct responses between native English and non-native English speakers ranges from 10% (US compared to AU; significant at p<0.05 using Fisher Exact test) to 20% (JP compared to US; significant at p<0.05 using Fisher Exact test). All other comparisons of the four countries used in this comparison (US, AU, JP and CH) were not significant.

As such, we report an upper and lower boundary on the frequency of wrong answers to these kinds of questions in the context of language. There are of course other factors involved in potential language differences than those addressed here. We suggest that the general scale of all answers to questions in this survey are very similar regardless of the language of respondents. To further account for possible language effects, we report averages for the frequency of particular answers as ranges to provide estimates of the potential impact of language on the overall conclusions of the study.

## Attitudes toward microbes and knowledge of antibiotics

We used the overall data set to examine the knowledge of respondents with respect to antibiotics (H2: Survey 1, Question 2 and Survey 2, Question 5). The Survey 1 question was "Which of these is an antibiotic? Select as many as you like!" with possible answers being Aspirin, Tylenol, Valium, Penicillin and Benadryl. The Survey 2 question (discussed above) included the same five compounds but added Azithromycin and Neosporin as possible answers. Both questions are objective with a well-defined correct answer (Survey 1, only penicillin is an antibiotic and in Survey 2 penicillin, Azithromycin and Neosporin [actually a combination of three antibiotics] are antibiotics).

In general, the surveys indicate that the public had difficulty identifying antibiotics by name. In other words, respondents had difficulty distinguishing antibiotics from other medicines. Because the respondents could give multiple answers to this question it is informative to report the percentage of respondents answering correctly that only penicillin is an antibiotic (rather than the number of respondents who picked penicillin along with other compounds) which was 49%. The overall percent of respondents answering incorrectly for aspirin, Tylenol, Benadryl and Valium (23%) as antibiotics are also shown (**Fig 2A**). Another way to look at the answers to these survey questions is simply by percentage of raw answers. These are recorded as overall averages for each compound as an antibiotic for Survey 1 in **Fig 2A** and are as follows: penicillin = 78%, aspirin = 28%, Tylenol = 20%, Benadryl = 23%, Valium = 24%. For Survey 2 the overall averages for identifying a compound as an antibiotic are aspirin = 21%, valium = 14%, Tylenol = 18%, penicillin = 63%, Benadryl = 63%, Neosporin = 33% and Azithromycin = 41% (**Fig 2B**).

In Survey 1 (**Fig 2A**), Europe (EU), Latin America (LA) and North America (NA) responders were significantly different from Africa (AF) and Asia (AS), but no other pairwise comparison of these geographic regions were significant. In Survey 2, EU, India (IND), NA and Pacific area (PAC) were significantly different from AF and AS, and NA is significantly different from EU, PAC and LA. The lower frequencies of identification of antibiotics by Asian, African, and Middle Eastern (ME) respondents could relate to lack of familiarity with the American brand names for these medicines, although generic and specific names were also provided in the questions and when possible, we included generic names for these compounds that might be more familiar to the foreign visitors.

Survey 2 asked the same question, including the same five compounds as well as another antibiotic (azithromycin) and an antibiotic mixture (Neosporin). Respondents misidentified the following compounds, aspirin, acetaminophen, Valium and Benadryl, as antibiotics at an average rate of 22% for each, similar to Survey 1. Only 26% of respondents identified both azithromycin and penicillin as antibiotics (**Fig 2B**). For North American (NA) respondents, only 21% could correctly identify all three antibiotics (including Neosporin which is a local brand name and an antibiotic mixture) as such. The summaries of individual answers to this question on Survey 2 are shown in **Fig 2C** where it is evident that each of the wrong answers (aspirin, Valium, Tylenol and Benadryl) are given on average over 20% of the time.

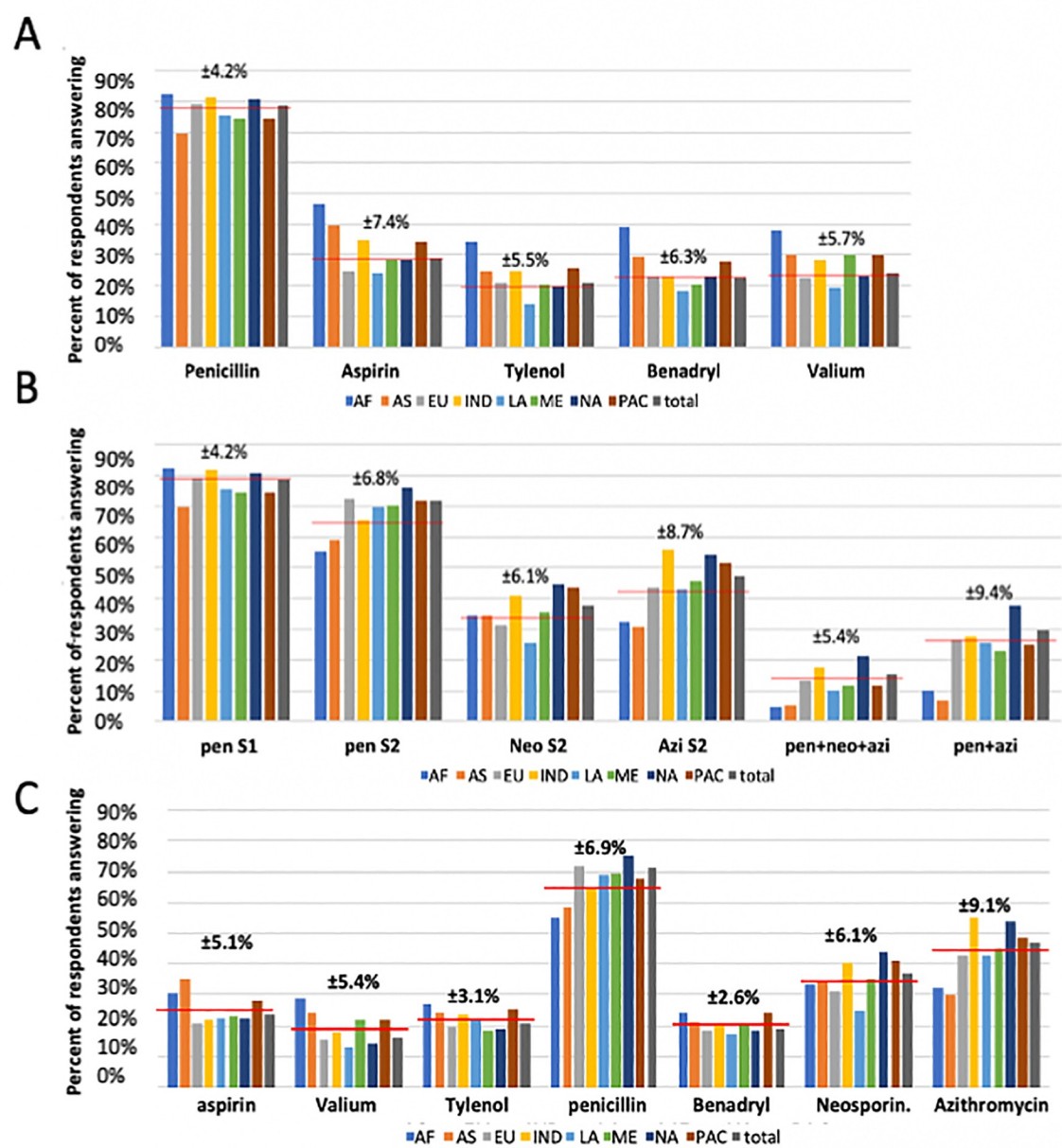

**Fig 2. Ability to identify antibiotics. (A).** The distribution of the indicated responses for Survey 1/Question 3. **(B)** The distribution of the indicated answers for Survey 2/Question 5 Both figures show the distribution by general geographic region. Geographic regions are Africa [AF], Asia [AS]. Europe [EU], India [IN], Latin and South America [LA], Middle East [ME], North America [NA] and Pacific Region [PAC]). PenS1 and PenS2 indicate the proportion of answers that identified penicillin as an antibiotic in Survey 1 and Survey 2 respectively. NeoS2 and AziS2 indicate identification in Survey 2 of Neosporin and Azithromycin as antibiotics. pen+neo+azi indicates the percent of answers that correctly identified penicillin, Neosporin and Azithromycin as antibiotics. Pen + Azi indicates the percent of answers that correctly identified penicillin and Azithromycin, as antibiotics. For each panel, the horizontal red lines indicate the overall average and the standard deviation is also shown above the bars. **(C)** Histograms summarizing percent of respondents answering the individual choices to "Which of the following are antibiotics?" for each of the possible choices. See https://sepakiosk.com/ for raw data. Abbreviations: NA = North America, ME = Middle East, LA = Latin America, IND = India, EU = Europe, AS = Asia, AF = Africa, PAC = Pacific region.

To approach H2 (**Table 1**; understanding the role of microbes in modern human life) we examined respondents' answers to Survey 1, Question 1 and Survey 2, Questions 1 and 2. In this way, we examined participant attitudes toward microbes using their preferences for

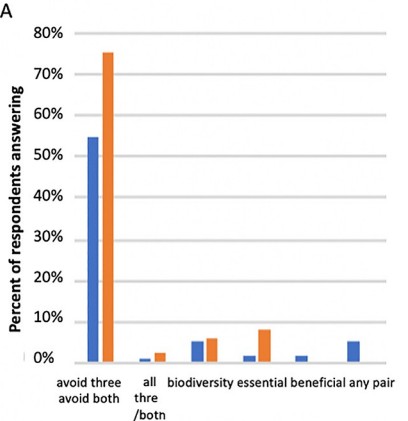
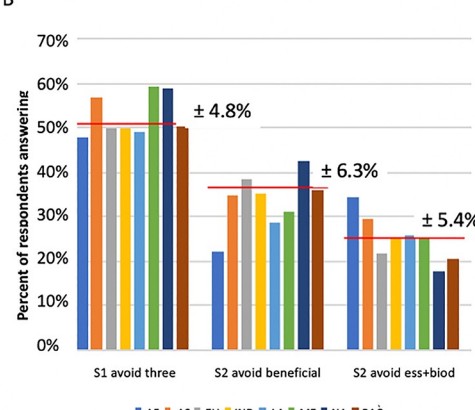

**Fig 3. Results of the comparison of combination and avoidance of terms (beneficial, essential and diverse) to describe microbes. (A)** Frequency in Survey 1 (blue) of the use of the words "beneficial","biodiversity" and "essential" in response to the question "Which words come to mind when you hear the word microbe" for survey 1 that also allowed respondents to answer "tiny", "germ" and "disease". *Avoid three* refers to Survey1/Question1 where respondents avoided all three positive terms (beneficial, biodiversity and essential). *All three* refers to the percent of respondents who used all three positive terms only in their answers. The frequency of similar answers to Survey 2/ Questions 1 and 2 are shown in orange. Biodiversity refers to the answer "There are many types of microbes" for Survey 2/Question1. Essential refers to the answer "Microbes are essential for life" for Survey 2/Question 1. Beneficial refers to the answer "Mostly beneficial" for Survey 2/Question 2. *Avoid both* refers to Survey 2/Question 1 where respondents answered both "There are many types of microbes" and "Microbes are essential for life" and both refers to respondents who gave those two answers to Survey 2/Question 1. **(B)** The left graph (S1 avoid three), refers to the percent of answers to Survey 1/Question 1 that avoided all three terms indicating positive aspects of microbes (beneficial, essential and diverse). The middle graph (S2 avoid beneficial), refers to the percent of answers to Survey 2/ Question 2 that avoided the term beneficial. The right graph (S2 avoid ess+biod), refers to the percent of answers to Survey 2/Question 1 that avoided the terms essential and biodiversity. For each panel, the red lines indicate the overall average and the standard deviation is shown. See https://sepakiosk.com/ for raw data.

particular descriptive words for microbes. In the first survey, we compared three common terms that often are used to describe microbes (tiny, disease, and germ) with three terms that describe positive aspects of microbes (essential, biodiversity, and beneficial) and asked "Which words come to mind when you hear the word microbe?" Although participants were able to list as many words as they wanted, responses were less likely to use the positive terms (**Fig 3A**). This finding is consistent with museum visitors having limited recognition of the benefits provided by microbes (**Fig 3A**).

In Survey 1, few of the respondents used any of the terms beneficial, essential, or biodiversity to describe microbes (**Fig 3B**). We found that 53% entirely avoided the use of the three terms, and only <1% used all three terms together (**Fig 3A**). Use of any pair of the three positive terms was uncommon (<5%) as were single usages of the three terms (all three single usages added to <10%).

To better understand the reluctance of respondents to use positive terms to describe microbes, in Survey 2, we used two different questions (**Table 2**); one focused on biodiversity and the essential nature of microbes (Question 1), and the other focused on the health benefits of microbes (Question 2). For Question 1, 75% of the responses avoid use of beneficial and diversity (i.e., many kinds of microbes), while only 3% used both terms (**Fig 3A**). For Question 2, 63% avoided the response that indicated microbes can be beneficial for health, showing lack of appreciation of beneficial roles in the majority (**Fig 3B**).

## Potential for comparing PAK trends across countries

Questions on these surveys can give us important information on the knowledge of antibiotics for a geographic region. We used the "accurate" counts for correct and incorrect answers to Survey 1, Question 2 and Survey 2, Question 5 to rank countries as to the ability of their respondents to correctly recognize antibiotic names. We used a cutoff for inclusion in these comparisons at N>95 for both surveys. **Table 4** shows the seventeen and nineteen countries with sample sizes greater than 95 for Survey 1 and Survey 2 respectively. The table shows that the United Kingdom ranks higher than any of the other countries using the approach for ranking that we employed. In addition, it is also clear that European countries rank higher than countries from elsewhere in the world at recognizing the names of antibiotics. Many of these are European Union countries that have benefit from programs on antibiotic resistance. We suggest that these analyses lean toward recognizing the United Kingdom as being perhaps the most informed population in the overall sample, indicating that we can reject H4 (**Table 1**; knowledge of antibiotics does not differ between countries).

The United Kingdom has recently instituted the second of its two five-year plans to educate their populace on antibiotic resistance [34,35]. While there have been efforts to educate about antibiotics in other European countries, such education has not been as intense as in the United Kingdom. To examine the efficacy of the British five year plans we posited a new hypothesis that British respondents would fare better at recognizing the names of antibiotics using the same questions discussed above ("Which of these is an antibiotic? Select as many as you like") for Survey 1 and Survey 2. We parsed a subset of the total dataset for the responses from 2017 for this question by sorting the data by country with 95 or more respondents.

**Table 4. Countries with >95 respondents answering for Survey 1 (S1) and Survey 2 (S2) used in the comparison of the United Kingdom to other countries.**

| Country | NS1 | RS1 | NS2 | RS2 |
|---|---|---|---|---|
| Argentina | 155 | 5 | 342 | 7 |
| Australia | 329 | 13 | 368 | 9 |
| Brazil | 275 | 9 | 418 | 12 |
| Canada | 562 | 14 | 739 | 14 |
| China | 326 | 15 | 407 | 15 |
| Columbia | 108 | 8 | 128 | 11 |
| France | 328 | 11 | 439 | 6 |
| Germany | 169 | 2 | 205 | 2 |
| India | 186 | 4 | 188 | 8 |
| Netherlands | | | 98 | 3 |
| New Zealand | | | 95 | 13 |
| Italy | 206 | 7 | 342 | 4 |
| Japan | 99 | 16 | 94 | 16 |
| Mexico | 140 | 10 | 128 | 13 |
| Russia | 530 | 17 | 761 | 17 |
| South Korea | 103 | 12 | 113 | 10 |
| Spain | 246 | 6 | 295 | 5 |
| UK | 352 | 1 | 424 | 1 |
| USA | 3877 | 3 | 4930 | 3 |

The number of respondents for survey 1 (NS1) and the number of respondents for Survey 2 (NS2) are given. The relative rank of a country for correct answers for Survey 1 (RS1) and Survey 2 (RS2) are also given. See text for how countries were ranked according to correct answers.

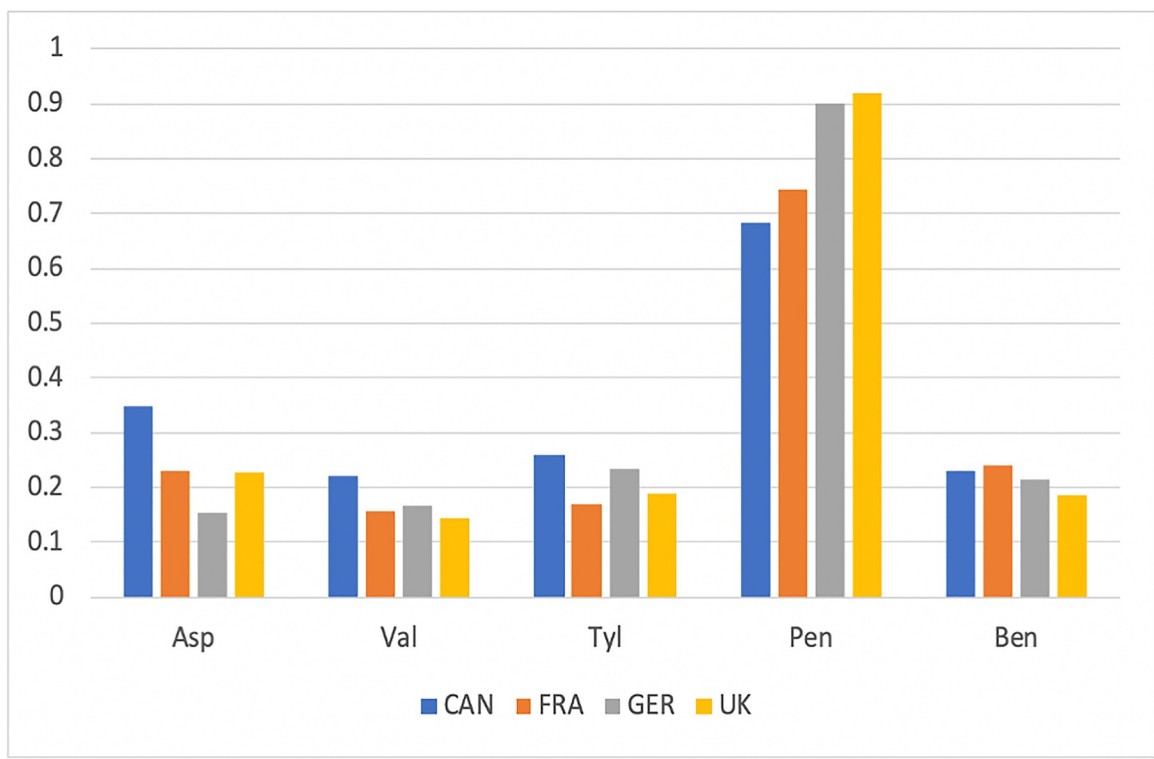

**Fig 4. Histogram showing the results of analysis of Survey 1 for the penicillin question discussed in the text for France, Germany, Canada and the United Kingdom.** The choices to the question "Which of these is an antibiotic? Select as many as you like!" were aspirin (Asp), Valium (Val), Tylenol (Tyl), penicillin (Pen) and Benadryl (Ben). Four countries were examined in this initial step–United Kingdom (UK), Germany (GER), France (FRA) and Canada (CA). The Y axis shows the percent of respondents giving the indicated answer.

We used the raw count of the times that correct and incorrect answers were given by a respondent for initial analyses. **Fig 4** shows the results for a subset of this comparison for Survey 1 for France, Germany, Canada and the United Kingdom. One striking result that is consistent with those described above for the overall analysis is that respondents from these countries answer that non-antibiotic medicinal compounds like aspirin, Tylenol, Benadryl and valium are antibiotics at a rate of about 22%. There are no significant differences between the UK answers and those from the other three countries for these non-antibiotics. However, with respect to our original hypothesis it is clear that the UK respondents do show up to a 30% higher frequency of identifying penicillin as an antibiotic relative to several other countries' respondents. The UK respondents show only slightly better identification of penicillin as an antibiotic over German respondents.

## Discussion

The results of this study are significant in two major areas– 1) Assessing public attitudes to microbes; 2) Testing knowledge of the public about antibiotics.

### Public attitudes to microbes

The results of the two surveys based on >22,000 respondents to three simple questions (Survey 1, Question 1; Survey 2, Question 1, Question 2) and subsequent test of H2, H3, H4 and H5 (**Table 1**) provide a snapshot of museum visitors' attitudes toward microbes and antibiotics.

To date very few survey studies have attempted to assess the attitude of the public to microbes on a as large a scale as here. The major focus in the literature on this topic is simply to state that we need to shore up the public's take on microbes [23,24,36,37]. The current study provides solid evidence that a good proportion of museum goers do not fully understand the essential and beneficial role of microbes in health. The surveys demonstrate a predominantly negative attitude by the public toward microbes as based on our analyses above as only 53% of the respondents entirely avoided the use of the three positive terms describing microbes and only 1% used all three positive terms together to describe microbes (**Fig 3**). Such a lack of understanding could impact the acceptance of public health measures based on the beneficial properties of microbes [25,36,38]. Understanding the role of microbes in the environment and in the human body is critical to the success of future public health measures to better steward antibiotic use [24,37,39]. We point out that the results reported here are for museum goers, a category which is predominately better informed than others [40–53]. Hence the results we show here should give an upper bound of the general public's attitudes.

## Public recognition of antibiotics

The literature on public knowledge of antibiotics is substantial. Our canvassing of the literature revealed nearly a hundred surveys attempting to address PAK in a large number of different countries (**S1 File**). The inclusion of questions on these surveys are for the most part consistent from survey to survey. For instance, the following question appeared in almost all of these published surveys "Answer TRUE/FALSE: Antibiotics are effective in killing viruses". Another kind of question that was relatively consistent across surveys in the literature involved correctly labeling a medicinal compound (Table 5) as an antibiotic such as "Answer TRUE/FALSE penicillin is an antibiotic" or Answer TRUE/FALSE aspirin is an antibiotic." We compiled and summarized results from these surveys relevant to these two questions (**S1 File** and **Table 5**). The first compilation shows the percentage, method of survey and country where the study queried about the effectiveness of antibiotics on flu, viruses or the common cold. The data presented in **S1 File** show the literature survey for this kind of question. The percentage of wrong answers for this question when "viruses" are inserted into the question is 20% to 90%. When "flu" is inserted the range of percentage of wrong answers is 30% to 93%. Interestingly there is a general trend of lower percentage of wrong answers from developed countries versus developing countries for both kinds of questions (**S1 File**). The kiosk survey results here are not directly comparable to the antibiotics/viruses/flu question discussed above as we did not directly ask this question in our surveys. However, the public gives wrong answers for antibiotic recognition in our study at about the same rate as they answer that flu and viruses can be overcome with antibiotics. An overall average of 54% of respondents from the literature survey answer incorrectly that viruses and flu can be controlled by antibiotics. For Survey 1, 51% of total respondents answer incorrectly that penicillin is not an antibiotic. For Survey 2, 45% of the overall respondents answered incorrectly that penicillin was not an antibiotic. These results are somewhat similar to those of Li et al., [54] for Chinese respondents. Li et al., [54] developed an antibiotic knowledge scale and included nearly 13,000 respondents in their study. They showed that 67% of their respondents "had poor antibiotic knowledge". While it is difficult to cross compare studies (discussed above) it is significant that the grand majority demonstrate over half of the public has poor understanding of antibiotics. This conclusion is discouraging as it demonstrates an overall and repeatable low knowledge level of the public concerning antibiotics.

Results of the literature survey for answers to the second category of questions on recognizing antibiotics is shown in **Table 5** and **S1 File**. These results are directly comparable to the

**Table 5. Representative surveys from the literature addressing antibiotic recognition.**

| Country | N | aspirin | paracet | Actual AB | Study |
|---|---|---|---|---|---|
| Bhutan | 692 | 21 | 32 | 43 | 61 |
| India (MS) | 382 | 37 | 35 | 90 | 62 |
| Italy | 1247 | 4 | 6 | 94 | 63 |
| Lebanon | 500 | 12.7 | 9.1 | 53.1 | 64 |
| Malaysia | 250 | 17 | | 39 | 65 |
| Malaysia | 383 | 33.4 | | 62 | 66 |
| Mexicao | 101 | 45 | 44 | 89 | 67 |
| Nepal | 220 | 15 | | 29 | 68 |
| NZ India | 130 | | | 67 | 69 |
| NZ Egypt | 102 | 28 | | 83 | 69 |
| NZ S. Korea | 104 | 41 | | 37 | 69 |
| Nigeria | 430 | 53 | 29 | 85 | 70 |
| Saudi (MS1) | 130 | 6 | 8 | 89 | 71 |
| Saudi (MS2) | 60 | 30 | 36 | 34 | 71 |
| Saudi (MS) | 347 | 29.7 | 16.7 | 26.8 | 72 |
| South Africa | 386 | 8 | | 64 | 73 |
| Tanzania | 292 | 29 | 28 | 58 | 74 |
| Trinidad | 753 | 9 | 10 | 83 | 75 |
| Turkey | 100 | 4 | 22 | 63 | 76 |
| Germany | 977 | 10 | 10 | 86 | 77 |
| Spain (MS) | 578 | 7 | | 85 | 78 |
| UK (MS) | 583 | 2 | | 96 | 79 |

The country is given in the "Country" column. (MS) indicates medical students. The New Zealand survey included people who migrated to New Zealand from India (NZ India), South Korea (NZ S. Korea) and Egypt (NZ Egypt). The sample size is given in the N column. The values under "aspirin" and "paracet" (Paracetemol) are frequencies of wrong answers (ie that aspirin or Paracetemol are antibiotics). The values under Actual AB are the frequencies of correct answers. The antibiotic most prominently used as an example was penicillin. The source of the data [55–73] is in the "Study" column.

results from our two kiosk surveys. When penicillin is the query word, the percent of correct answers in the literature survey ranges from 28% to 95%. When aspirin is the query word, the percent of correct answers in the literature survey ranges from 47% to 96%. These ranges can be compared to the ranges we infer from Survey 1 (Penicillin: 70% to 83%; average 78%+/- 4.2%; Aspirin: 24% to 46%; average 29% +/- 7.4%) and from Survey 2 (Penicillin: 70% to 83%; average 78%+/- 4.2%; Aspirin: 24% to 46%; average 29% +/- 7.4%). While the ranges in the literature are large there is general overlap and the overall conclusion that there is widespread poor understanding about antibiotics is sustained.

## Potential for comparing PAK trends; the UK versus other countries

Table 4 shows the rankings of antibiotic recognition for several countries with sample sizes in our study with N>95. It is clear that the United Kingdom respondents fare better than any of the other countries with adequate sample size. The rankings appear to be consistent from Survey 1 to Survey 2, with the top three ranked countries the same in both surveys and the bottom four ranked countries the same in both surveys. In addition, the literature survey ranked the UK first, with EU countries Spain, France, Germany and Italy also ranked in the top six countries which is very similar to our kiosk surveys. In general, our surveys, like the relatively large literature on antibiotic recognition, clearly show that the public has difficulty identifying antibiotics by name and distinguishing them from other medicines.

## Limitations

The two surveys over a two-year period (2016–2018) were conducted using a kiosk installed in the American Museum of Natural History. While kiosk surveys are commonly used to evaluate products for businesses they have in limited cases been used in health education and public knowledge assessment projects [74–76]. The utility of kiosks as a survey tool is not well known, but it is reasonable to suggest that kiosk interaction is mechanically similar to online surveying which has been considered very useful [77–79].

Another related issue was the location of the kiosk. While we placed the kiosk as far away as possible from any exhibits in the AMNH focused on microbes, there was still the potential for on-site learning of concepts immediately prior to the survey. Such on-site learning would inflate the results in the same direction as using museum goers as subjects and would inflate the degree of understanding of museum goers surveyed in this study.

The makeup of the participants (museum goers) in our surveys is complicated by being conducted in a science museum. Several studies have examined the level of PAK in museum goers for many different topics [40–53] and the consensus is that museum goers are better informed about a wide range of scientific topics than non-museum goers. In addition, the comparison of different geographic regions here might not be entirely valid for calling this study a global study. We point out that the participants in our study are probably at the more knowledgeable end of the general public PAK spectrum. Our results can therefore be interpreted as a best-case scenario with respect to the level of knowledge in regard to the survey questions. More than likely the level of knowledge of the general population is lower than what we observe here.

Any study using a survey is based on the reliability and validity of the questions posed. While no formal analysis of these survey parameters was accomplished prior to the survey we suggest that our pre survey exploration addressed some of the validity concerns that surveys face. We also suggest that the consistency of similar answers from the survey done in 2017 with those from 2018 indicate a degree of reliability of those survey questions. Finally, we point out that several of the questions in our survey have been posed before in other surveys on antibiotics and microbial issues, which also suggesting a degree of validity of this study.

A final limitation of our study concerns the language we used to distribute the surveys— English. We examined this problem in some detail (S3 File), but a language barrier might be responsible for some of the patterns we observe when we compare results across different geographic areas of the globe. A problem specific to language usage concerns using brand names in the antibiotic questions in the survey. This complication would deflate our observation of the degree of understanding of museum goers surveyed in this study.

## Conclusion and education policy issues

Together, the results on peoples' attitudes to microbes and knowledge of antibiotics indicate a substantial lack of familiarity with both, which implies a need for better education of the general public and of museum goers about these subjects. With antibiotic resistance increasing [16], and the most recent global Covid 19 pandemic [17–19] we need to understand better what people know and hence assess better what they need to know both locally and globally. From the perspective of a natural history museum or a science center knowing where to focus efforts is a first step in developing functional and effective programs and exhibitions.

The current study was part of a larger initiative started at the AMNH in 2015 to educate the public about microbes and issues related to microbes that included two exhibitions on the microbiome and Science Café programming events as well. The surveys were designed by the museum's educators, exhibition staff and scientific advisory panel to probe the museum goer's

knowledge of certain issues connected to microbial diversity and microbial life. The data from the surveys can inform the museum educators and exhibition staff as to what level materials developed by the outreach and informal education arms of the museum should be targeted at. The AMNH uses such information to create a system for gauging audience interest in particular topics, as well as identify gaps in audience knowledge between cutting-edge research on biodiversity and health and public perceptions of those intersections and their implications. The current results indicate that in developing future programming.

We also suggest that the analysis here can be disseminated for the benefit of other formal and informal science institutions, providing them with methodology and data they can use for their own programming. This survey of public knowledge will be particularly valuable in influencing critical conversations on national science education and science policy, and dissemination efforts aimed to reach these relevant audiences. These results then can help guide the design of specific education programs. Public education programs about microbes often start with the somewhat sophisticated topic of antimicrobial resistance. We demonstrate a lack of general knowledge about microbes in the museum going public that suggest museum educators and perhaps even public health educators should reassess the level at which information about microbes is initially presented [23,24].

## Supporting information

**S1 File. Comparison of antibiotic recognition in the literature.** The literature on public knowledge of antibiotics is substantial. The comparisons span a large number of countries and demgraphies.
(DOCX)

**S2 File. A description of the kiosk used in the surveys with information for access to survey questions, data and information on the AMNH-SEPA project.**
(DOCX)

**S3 File. Analysis of the impact of language differences on survey responses.**
(DOCX)

## Author Contributions

**Conceptualization:** Julia Zichello, Preeti Gupta, Monique Scott, Bella Desai, Ruth Cohen, Lauri Halderman, Ana Porzecanski, Paul J. Planet, Yael Wyner, Martin Blaser, Robert Burk, Judy Diamond, Jennifer Borland, Rob DeSalle.

**Data curation:** Lauri Halderman, Jennifer Borland, Rob DeSalle.

**Formal analysis:** Susan Perkins, Yael Wyner, Martin Blaser, Jennifer Borland, Rob DeSalle.

**Funding acquisition:** Preeti Gupta, Monique Scott, Ruth Cohen, Rob DeSalle.

**Investigation:** Julia Zichello, Bella Desai, Lauri Halderman, Susan Perkins, Ana Porzecanski, Paul J. Planet, Yael Wyner, Martin Blaser, Robert Burk, Judy Diamond, Jennifer Borland, Rob DeSalle.

**Methodology:** Bella Desai, Lauri Halderman, Ana Porzecanski, Paul J. Planet, Yael Wyner, Robert Burk, Judy Diamond, Jennifer Borland, Rob DeSalle.

**Project administration:** Julia Zichello, Preeti Gupta.

**Resources:** Monique Scott, Lauri Halderman, Yael Wyner, Rod Kennett.

**Software:** Lauri Halderman.

**Validation:** Susan Perkins, Paul J. Planet, Yael Wyner, Martin Blaser, Robert Burk, Judy Diamond, Rod Kennett, Rob DeSalle.

**Visualization:** Rob DeSalle.

**Writing – original draft:** Rod Kennett, Rob DeSalle.

**Writing – review & editing:** Rod Kennett.

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
