## [Decision Letter · Decision Letter 0]

8 Mar 2021

PONE-D-21-01340

A Natural History Museum Visitor Survey of Perception, Attitude and Knowledge (PAK) of Microbes and Antibiotics

PLOS ONE

Dear Dr. DeSalle,

Thank you for submitting your manuscript to PLOS ONE. After careful consideration, we feel that it has merit but does not fully meet PLOS ONE’s publication criteria as it currently stands. Therefore, we invite you to submit a revised version of the manuscript that addresses the points raised during the review process.

Please respond to these requests, especially those by reviewers 1 and 3. Please note that you need not address formatting requests at this stage. Moreover, while addressing requests about the novelty and implications of your research, please ensure that this discussion is limited to items directly impacted by the findings. The data presented in the manuscript must support the conclusions drawn. Authors may discuss possible implications for their results as long as these are clearly identified as hypotheses instead of conclusions.

We look forward to receiving your revised manuscript.

Sincerely,

Yann Benetreau, PhD

Senior Editor, *PLOS ONE*

Journal Requirements:

Additional Editor Comments (if provided):

Reviewers' comments:

Reviewer's Responses to Questions

**Comments to the Author**

1. Is the manuscript technically sound, and do the data support the conclusions?

Reviewer #1: No

Reviewer #2: Yes

Reviewer #3: No

2. Has the statistical analysis been performed appropriately and rigorously? 

Reviewer #1: I Don't Know

Reviewer #2: Yes

Reviewer #3: No

3. Have the authors made all data underlying the findings in their manuscript fully available?

Reviewer #1: Yes

Reviewer #2: Yes

Reviewer #3: Yes

4. Is the manuscript presented in an intelligible fashion and written in standard English?

Reviewer #1: Yes

Reviewer #2: Yes

Reviewer #3: No

5. Review Comments to the Author

Reviewer #1: # Overview

This is a descriptive study that aims to "assess public knowledge of antibiotics and public attitudes toward microbes in museum goers." While the data used in this report are from an impressively large sample, the explication of the concepts and study rationale are lacking and I cannot recommend publication.

My comments below address the shortcomings of the manuscript and I hope the authors find these helpful in revising their work.

# Concept explication and study rationale

Overall, the concepts of interest are not clearly defined in this study. Although the authors purport to assess public attitudes, there are no hypotheses about visitors' attitudes toward microbes. The hypotheses in Table 1 address museum attendees' factual knowledge about microbes and antibiotics, not attitudes.

I recommend the authors revisit their conceptual definitions based on the existing operationalizations (since data have already been collected). They could also do one of two things: (i) restructure the report to focus solely on factual knowledge; or (ii) pose hypotheses and/or research questions about attitudes.

If the authors opt for (ii), the survey questions that are valid measures of attitudes are the ones about hand sanitizer (Q3 in Survey 1 and Q4 in Survey 2). The authors could also use data from the question asking about risks and benefits as an attitudinal measure. Although the authors identify this question as asking about public confidence in their knowledge, this is more appropriately conceptualized as a question designed to assess how familiar visitors *think they are* with risks and benefits of antibiotics. That said, this question is double-barreled--respondents are asked about their perceived familiarity with two things (risks *and* benefits). How, for example, would a visitor respond to this question if they felt familiar with one and not the other?

I also recommend that the authors conduct and include a literature review for each concept of interest, whether it is knowledge/understanding or attitudes. This will help tie the existing hypotheses (and any additional ones) together--H1 and H4 seem quite distinct from H2 and H3. The former are about differences between groups of respondents while the latter collapse these groups. A stronger rationale that situates these hypotheses in the existing literature is needed.

Regarding the study rationale, the authors appear to have overlooked a large literature on the knowledge deficit model. This is especially relevant since the existing hypotheses are about knowledge--what is the rationale for studying levels of knowledge among museum visitors given the substantial extant literature that makes clear factual knowledge is not the only predictor of people's attitudes toward a given scientific issue?

# Other comments

- There remain some unsupported claims in the manuscript (e.g., "We point out that the results reporteed here are for museum goers, a category which is predominately better informed than others."). Please include page and/or line numbers for ease of review.

- There remain typos in the manuscript (e.g., "guage" [Table 2]).

- Q4 in Survey 2: "3 Hand sanitizer is a convenient e alternative to hand washing." Is the "e" a typo? Was this in the questionnaire?

- The font size of the latter half of the Conclusion appears different, though this might be a product of the manuscript submission system converting the document into a PDF.

- "The validity of kiosks as a survey tool is not well known, but it is reasonable to suggest that kiosk interaction is similar to online surveying..." Why is this reasonable to suggest? Please offer an explanation. Kiosk surveys employ convenience sampling while online surveys can be conducted through a variety of sampling methods.

- In the Results, the authors state, "Respondents misidentified the following compounds, aspirin, acetaminophen, Valium and Benadryl, as antibiotics at an average rate of 22% for each, similar to Survey 1." Please explain how a rate is calculated from cross-sectional survey responses (I assume these are cross-sectional data even though the fielding period is relatively long).

- Please include more information about data analysis in your Method section. For example, there is little information about how the impact of native and non-native English speakers was assessed.

- The authors claim that "analysis of language patterns also revealed greater variation of answers related to medical compound names..." How was this accomplished? Or perhaps the term "language patterns" is inaccurate since the data do not necessarily speak to this. Given the closed-ended nature of the kiosk survey, how can these data be used to assess "language patterns?"

- How do the authors know, for example, that the differences in proportions of correct responses in data from different countries are not due to a spurious variable? This analysis does not necessarily address differences between native and non-native English speakers. Instead, any differences in the comparisons may be a result of broader, cultural factors.

- The hypotheses refer to "public" knowledge and understanding. Yet, the sample is not a representative sample of the global public. I recommend the authors not use this term and instead refer to units in the sample as "visitors" or "respondents." To refer to the sample as a "public" one is misleading.

- Please defend the choice of statistical test for examining the impact of non-native vs. native English speakers. The Fisher exact test is typically used for small sample sizes. Given the sample size, it seems more appropriate to use a chi-square test.

- I believe there are typos in Table 2, specifically in H2 and H4. I think the authors meant to refer to Q4 in Survey 2, not Q3, which is about hand sanitizer, according to Table 1.

- In general, there needs to be more detail of the analysis included in the manuscript. For example, against what standard did the authors compare H2? What is the threshold after which we can say that visitors "understand" the role of microbes? This question is also valid for H3. For H1 and H4, please specify the statistical tests employed in the Method section.

Reviewer #2: This is a very interesting paper, which provides information on how to promote proper public awareness of antibiotics in the context of the Covid-19 pandemic. I have some suggestions to improve the manuscript.

1. The format of the citation should meet the requirements of the journal.

2. Why were visitors to the Natural History Museum chosen as the study object?

3. To what extent can museum visitors represent lay public?

4. In the Introduction section, it would be better to describe the results of some current studies. Please include and discuss the following reference https://pubmed.ncbi.nlm.nih.gov/33022320/

5. So far, the survey design section is too piecemeal, please make it more concise.

6. The authors mentioned that in order to assess the influence of language on the survey answers, they chose two questions to measure? What are the two specific questions and why do you choose them?

7. “We point out that the results reported here are for museum goers, a category

which is predominately better informed than others.” Need references for this statement.

8. The first paragraph of the discussion section describes not the results of your research but the purpose of your research.

9. In the Discussion section. the authors should add 1 to 2 paragraphs to talk about the policy implications.

10. I don't think it's necessary to cite references in the Limitations and Conclusions section, which should have been discussed earlier.

Reviewer #3: The study is an interesting and novel one however there are a lot of problems with the paper as it is. It is very difficult to follow- there are a lot of information not mentioned in the correct place (for eg results mentioned in the introduction, or discussion sections mentioned in the methods etc). Although it is interesting there are also several methodological flaws and limitations to the work. And your discussion and conclusion are generalised although this shouldn't be the case. Introduction:

In the last paragraph you describe some limitations and the overall outcomes/results of the study which doesn’t need to be there yet. This is better fit in the discussion and conclusion. Ensure that you add the aim and/or objectives of the study at the end of your introduction instead.

Methods:

My understanding was that you conducted 2 separate surveys with different questions (aside from the 3 identical demographic questions) and then you have decided to only present that data for some of the questions in survey 1 and some for survey 2? There is no explanation to why this approach was taken rather than presenting and analysing the data for survey 1 altogether particularly as you mention that some of the questions in survey 2 may have been further modified based on survey 1 (in which case survey 1 maybe considered a pilot survey). There is no information on how the different surveys were analysed- whether together or separately? Did you group all the demographic data for survey 1 and 2? No meaningful statistical analysis to indicate correlation between languages/ knowledge. No information on whether the validity and reliability of the questionnaires has been assessed either! Did the authors consider a specific sample size (calculating a sample size) for their survey?

Results:

You mention using the Fisher Exact test in the results section- such information should be included in the methods section as part of the data analysis. What is the rationale for using a cut-off point of N>95?

Table 5: unfortunately, I do not understand what is meant by NZ Egypt, NZ India etc the explanation is unclear.

Figure legends should not include an explanation of the result- but just an explanation of what the figures show- the explanation and analysis should come within the text not the figure legend.

The discussion brings out a few interesting threads however the overall conclusion and recommendations are very generalised when this shouldn’t be the case because of the diversity of the participants as well as the limitations mentioned. Perhaps it would be more realistic to have a recommendation based on the museum’s educational programmes/ sections etc to address this identified lack of knowledge in the first instance.

6. PLOS authors have the option to publish the peer review history of their article (what does this mean?). If published, this will include your full peer review and any attached files.

Reviewer #1: No

Reviewer #2: No

Reviewer #3: No

---

## [Author Response · Author response to Decision Letter 0]

29 Apr 2021

Reviewer Comments and Response

Reviewer #1: # Overview

This is a descriptive study that aims to "assess public knowledge of antibiotics and public attitudes toward microbes in museum goers." While the data used in this report are from an impressively large sample, the explication of the concepts and study rationale are lacking and I cannot recommend publication.

My comments below address the shortcomings of the manuscript and I hope the authors find these helpful in revising their work.

# Concept explication and study rationale

Overall, the concepts of interest are not clearly defined in this study. Although the authors purport to assess public attitudes, there are no hypotheses about visitors' attitudes toward microbes. The hypotheses in Table 1 address museum attendees' factual knowledge about microbes and antibiotics, not attitudes.

To alleviate the reviewer’s criticism that our concepts of interest are vague we have added the following at the beginning of the second paragraph in the introduction. While the initial surveys we conducted for this study addressed a broad array of microbe related topics, here we examine specifically, and in detail, two aspects of public knowledge about microbes. The first addresses the public’s familiarity with antibiotics and the second focuses on how the public views microbes. The former - public knowledge about antibiotics - has been extensively addressed with copious published surveys (Supplemental File 1). The lattr concerns general attitudes or familiarity toward microbes, a topic which has not been well-surveyed. While there has been some research in the area of PAK about microbes in general 

We have also attempted to address this critique by adding more specific hypotheses to Table 1. Entitled “Hypotheses and tests of public knowledge and attitude to antibiotics and microbes”, this table in the original submitted manuscript was developed to address criticism like the one made by reviewer 1. 

The two hypotheses are

H2. The public perceives microbes as dangerous and pathogenic 

H3. The public perceives microbes as having no beneficial qualities 

These two hypotheses in the reviewed draft were grouped under one hypothesis

H2. The public understands the role of microbes in modern life

I recommend the authors revisit their conceptual definitions based on the existing operationalizations (since data have already been collected). They could also do one of two things: (i) restructure the report to focus solely on factual knowledge; or (ii) pose hypotheses and/or research questions about attitudes.

I think reformulating the hypothesis H2 in Table 1 goes a long way to do this. Also I think the reviewer was under the impression that we were addressing other topics in the survey than the two we actually did examine. So for Table 2 we have removed the questions we asked on the survey that were not discussed or analyzed in the current version of the paper. We hope it is clearer now that we don’t distract the reader with these other questions about probiotics and hand sanitizer. 

If the authors opt for (ii), the survey questions that are valid measures of attitudes are the ones about hand sanitizer (Q3 in Survey 1 and Q4 in Survey 2). The authors could also use data from the question asking about risks and benefits as an attitudinal measure. Although the authors identify this question as asking about public confidence in their knowledge, this is more appropriately conceptualized as a question designed to assess how familiar visitors *think they are* with risks and benefits of antibiotics. That said, this question is double-barreled--respondents are asked about their perceived familiarity with two things (risks *and* benefits). How, for example, would a visitor respond to this question if they felt familiar with one and not the other?

Again, we do not analyze hand sanitizer attitudes in this paper. We focus in this manuscript on two simple aspects of our survey – 1. Knowledge of antibiotics and 2. Basic perception and knowledge of microbes. While interesting we also do not approach the self-assessed attitudes of people in this paper. 

I also recommend that the authors conduct and include a literature review for each concept of interest, whether it is knowledge/understanding or attitudes. This will help tie the existing hypotheses (and any additional ones) together--H1 and H4 seem quite distinct from H2 and H3. The former are about differences between groups of respondents while the latter collapse these groups. A stronger rationale that situates these hypotheses in the existing literature is needed.

We did this. Table 5 is the result of extensive literature surveys of publications concerning different countries and their knowledge of antibiotics. They are based on a large number of publications. With respect to the second aspect of the paper concerning public attitude toward microbes, there are very few papers in the literature except for those saying that we need to address the issue so that people are more familiar with microbial topics. Our study was meant to be an initial exploration of this topic. I can go through and list references addressing antibiotic knowledge (there over 100 of these survey type papers in the literature) and also go through and list the references that address public perception of microbes (as stated in the manuscript there are many fewer of this kind of paper) if you like. But reference to these papers is copious throughout the manuscript and in the tables.

Regarding the study rationale, the authors appear to have overlooked a large literature on the knowledge deficit model. This is especially relevant since the existing hypotheses are about knowledge--what is the rationale for studying levels of knowledge among museum visitors given the substantial extant literature that makes clear factual knowledge is not the only predictor of people's attitudes toward a given scientific issue?

We looked into the knowledge deficit model literature and we do not feel it is terribly relevant for this study. In addition, we agree that clear factual knowledge of an issue or a topic does not mean the person will have a specific kind of attitude or perception of that topic. Of course religious, political, economic and other issues are involved. This study. Does not attempt to address that at all, but rather we hoped to establish a baseline for our study subjects for two very basic ideas about microbes.

# Other comments

- There remain some unsupported claims in the manuscript (e.g., "We point out that the results reported here are for museum goers, a category which is predominately better informed than others."). 

I am not sure I understand this comment, but if the reviewer is saying that the statement that “museum goers, a category which is predominately better informed than others” is unsubstantiated or unsupported then as did in the paper we refer the reviewer to 

Schneider, 2017; Gunther 1999, Kirchberg, 1996; Dickenson, 1992; Marty, 2007; Hein, 2006); Falk and Needham, 2011, 2013; Falk and Dierking, 2013; Şentürk and Özdemir, 2014; Falk et al., 2014, 2016; Suter, 2014; Larry, 2014; Hammerness et al., 2016.

All of these are cited in the paper, all lean toward the conclusion that museum goers are in general better informed about a wide range of scientific topics.

Please include page and/or line numbers for ease of review.

- There remain typos in the manuscript (e.g., "guage" [Table 2]).

- Q4 in Survey 2: "3 Hand sanitizer is a convenient e alternative to hand washing." Is the "e" a typo? Was this in the questionnaire?

- The font size of the latter half of the Conclusion appears different, though this might be a product of the manuscript submission system converting the document into a PDF.

These have all been fixed(see track changes)

- "The validity of kiosks as a survey tool is not well known, but it is reasonable to suggest that kiosk interaction is similar to online surveying..." Why is this reasonable to suggest? Please offer an explanation. Kiosk surveys employ convenience sampling while online surveys can be conducted through a variety of sampling methods.

We have changed the wording of this sentence. What we wanted to say is that mechanically kiosks and online surveys are similar and that their utility levels are similar. The new sentence read:

The utility of kiosks as a survey tool is not well known, but it is reasonable to suggest that kiosk interaction is mechanically similar to online surveying which has been considered very useful (Evans and Mathur, 2018; Ha et al., 2020; Dodemaide et al., 2020).

- In the Results, the authors state, "Respondents misidentified the following compounds, aspirin, acetaminophen, Valium and Benadryl, as antibiotics at an average rate of 22% for each, similar to Survey 1." Please explain how a rate is calculated from cross-sectional survey responses (I assume these are cross-sectional data even though the fielding period is relatively long).

This rate is a simple ratio of the number of responses that gave a particular answer (say “Valium”) divided by the total. 

- Please include more information about data analysis in your Method section. For example, there is little information about how the impact of native and non-native English speakers was assessed.

There is an entire supplemental file - Supplemental File 3 - where this is described. - 

- The authors claim that "analysis of language patterns also revealed greater variation of answers related to medical compound names..." How was this accomplished? Or perhaps the term "language patterns" is inaccurate since the data do not necessarily speak to this. Given the closed-ended nature of the kiosk survey, how can these data be used to assess "language patterns?"

The reviewer is correct. We meant here the pattern in the results not in language itself. We correct this in the text.

The sentence now reads:

However, analysis of language usage also revealed greater variation of answers related to medical compound names, but less with choice of descriptive words (like beneficial or essential).

- How do the authors know, for example, that the differences in proportions of correct responses in data from different countries are not due to a spurious variable? This analysis does not necessarily address differences between native and non-native English speakers. Instead, any differences in the comparisons may be a result of broader, cultural factors.

We can’t answer this question with the current study. We had hope Supplemental File 3 would answer many of these questions. This file was originally in the paper itself, but a reviewer requested we remove it and make it a supplemental file. In this file we discuss the differences between people with different languages, and offer some explanations for why we observe the pattern. However, we point out that the results for non-English speakers is very similar to English speaking and the differences we point out in the manuscript are clear.

- The hypotheses refer to "public" knowledge and understanding. Yet, the sample is not a representative sample of the global public. I recommend the authors not use this term and instead refer to units in the sample as "visitors" or "respondents." To refer to the sample as a "public" one is misleading.

The first round of review made this same comment, and we removed reference to the study as being “global”.

There is one reference in the current version where we admit that even though there are people included in the survey from all over the world, that we do not claim this study to be “global” but rather the study can be interpreted as indicative of the best case scenario

We point out that the participants in our study are probably at the more knowledgeable end of the general public PAK spectrum. Our results can therefore be interpreted as a best-case scenario with respect to the level of knowledge in regard to the survey questions.

- Please defend the choice of statistical test for examining the impact of non-native vs. native English speakers. The Fisher exact test is typically used for small sample sizes. Given the sample size, it seems more appropriate to use a chi-square test.

We have redone all tests where appropriate with Chi-Square statistics. The following paragraph in Supplemental File 3 summarizes these tests:

The only correct answer was penicillin alone; if a respondent indicated that other medicines were antibiotics, then their response was considered to be incorrect, even when penicillin was selected. (AU=Australia, US=United States of America, JP=Japan and CH=China). The disparity in correct responses between native English and non-native English speakers ranges from 10% (US compared to AU; p = 2E-5 using Chi Square Test) to 20% (JP compared to US; p = 0.0094 using Chi Square test; all other pairwise comparisons are p > 0.05 using Chi Square test). These analyses also reveal greater variation related to more technical terms like compound names, but less with usage of descriptive words (Supplemental Figure 1.1). 

- I believe there are typos in Table 2, specifically in H2 and H4. I think the authors meant to refer to Q4 in Survey 2, not Q3, which is about hand sanitizer, according to Table 1.

This table has been fixed. The reviewer is absolutely correct that the question numbers were jumbled. The question numbers are now easier to deal with as we have removed the hand sanitizer and probiotic questions (which we did not analyze or comment on for this paper).

- In general, there needs to be more detail of the analysis included in the manuscript. For example, against what standard did the authors compare H2? What is the threshold after which we can say that visitors "understand" the role of microbes? This question is also valid for H3. For H1 and H4, please specify the statistical tests employed in the Method section.

This comment is very interesting and will be employed in future kiosk survey methodology. One way to answer the question is through comparisons which we tried to do.

Reviewer #2: This is a very interesting paper, which provides information on how to promote proper public awareness of antibiotics in the context of the Covid-19 pandemic. I have some suggestions to improve the manuscript.

1. The format of the citation should meet the requirements of the journal.

We have worked on this and in the final manuscript the references will be uniformly formatted.

2. Why were visitors to the Natural History Museum chosen as the study object?

This is explained on the SEPA kiosk website and in Supplemental File 2. Basically, the answer is that we were given a grant by the NIH SEPA program to do educational public programming around microbes and the microbiome. One of the parts of the proposal was to assess public knowledge of various aspects of microbes and since we have used kiosk methods in the past to do this we decided that an in-house kiosk was appropriate (see Supplemental File 2). 

3. To what extent can museum visitors represent lay public?

Museum goers don’t represent the lay public. Several studies all cited in the manuscript have shown that museum goers are in general better informed about a wide array of scientific topics. (see above). I think we are very clear about that in the paper.

4. In the Introduction section, it would be better to describe the results of some current studies. Please include and discuss the following reference https://pubmed.ncbi.nlm.nih.gov/33022320/

This is an important paper and has been incorporated into the text and properly referenced.

5. So far, the survey design section is too piecemeal, please make it more concise.

We hope it is more concise now that we have removed reference to probiotics and hand sanitizer. We have also honed the hypotheses listed in Table 1, and have added a limitations section. We hope these changes have made the paper easier to read and more concise.

6. The authors mentioned that in order to assess the influence of language on the survey answers, they chose two questions to measure? What are the two specific questions and why do you choose them?

The language impact is discussed in detail in Supplemental File 3. As stated above we were asked by a reviewer to move it to the supplemental info during first round of review.

7. “We point out that the results reported here are for museum goers, a category

which is predominately better informed than others.” Need references for this statement.

I am not sure why two reviewers missed these references in the text, but they are there

Schneider, 2017; Gunther 1999, Kirchberg, 1996; Dickenson, 1992; Marty, 2007; Hein, 2006); Falk and Needham, 2011, 2013; Falk and Dierking, 2013; Şentürk and Özdemir, 2014; Falk et al., 2014, 2016; Suter, 2014; Larry, 2014; Hammerness et al., 2016.

8. The first paragraph of the discussion section describes not the results of your research but the purpose of your research.

Not sure what we can do here. I disagree slightly with the reviewer as the results are discussed in this first paragraph of discussion. No changes were made to address this criticism.

9. In the Discussion section. the authors should add 1 to 2 paragraphs to talk about the policy implications.

This is an interesting comment and similar to R3’s last comment and so we have added a section on policy changes or alterations to teaching strategy at the museum. See R3 responses for the language in this added section.

10. I don't think it's necessary to cite references in the Limitations and Conclusions section, which should have been discussed earlier.

I am not sure what to do with this comment either. We listed the limitations of the study in the discussion because we felt it necessary to make clear that certain inferences from the study should be regarded with caution. Placement of the limitations at the end of the discussion we felt was appropriate. As for references in these sections, we think it is important to have the ones we cite in this section and in the conclusion.

Reviewer #3: The study is an interesting and novel one however there are a lot of problems with the paper as it is. It is very difficult to follow- there are a lot of information not mentioned in the correct place (for eg results mentioned in the introduction, or discussion sections mentioned in the methods etc). 

We went through each section of the paper and tried to streamline with respect to the organizational issues raised by the reviewer. The Introduction is purged of results now. The Materials and Methods section is now half as long as it was in previous draft. The Results Discussion sections are now tighter too. 

Although it is interesting there are also several methodological flaws and limitations to the work. And your discussion and conclusion are generalised although this shouldn't be the case. 

We tried very hard not to generalize in the discussion and indeed added a limitations section, and for each major result we show we pointed out that the results are unique here for museum goers. 

Introduction:

In the last paragraph you describe some limitations and the overall outcomes/results of the study which doesn’t need to be there yet. This is better fit in the discussion and conclusion. Ensure that you add the aim and/or objectives of the study at the end of your introduction instead.

Reference to limitations in intro has been removed. And we have added an explicit statement about what the study objectives were. To do tjis we added this paragraph

In this study, we examine several hypotheses (Table 1) relevant to people’s PAK of microbes and antibiotics. Several sub-hypotheses can be generated from the four general ones listed in Table 1. Our approach to testing these hypotheses is also outlined in Table 1 and such tests aid us in approaching the objectives of the study. These objectives are twofold; first to assess the museum goer’s knowledge of and familiarity with microbes. Second, we wanted to add to the burgeoning literature concerning general knowledge of antibiotics.

Methods:

My understanding was that you conducted 2 separate surveys with different questions (aside from the 3 identical demographic questions) and then you have decided to only present that data for some of the questions in survey 1 and some for survey 2? 

Yes that is correct. We have removed the questions on probiotics and hand washing as they were more about personal habits. We think this approach is clearer now that we have removed specific reference to the questions about hand-sanitizer and probiotics. 

There is no explanation to why this approach was taken rather than presenting and analysing the data for survey 1 altogether particularly as you mention that some of the questions in survey 2 may have been further modified based on survey 1 (in which case survey 1 maybe considered a pilot survey). 

We added the following to justify focusing in on the questions about microbes and antibiotics:

While the initial surveys we conducted for this study addressed a broad array of microbe related topics, here we examine specifically, and in detail, two aspects of public knowledge about microbes. The first addresses the public’s familiarity with antibiotics and the second focuses on how the public views microbes. The former - public knowledge about antibiotics - has been extensively addressed with copious published surveys (Supplemental File 1). The latter concerns general attitudes or familiarity toward microbes, a topic which has not been well-surveyed.

There is no information on how the different surveys were analysed- whether together or separately? 

We have tried to flesh out the materials and methods and think that it is very clear now that the two surveys were analyzed separately. They pretty much had to be because there were different questions on the two surveys in the two major knowledge areas we approached. We also point out that answers to questions on the first survey sometimes guided questions on the second survey. The bottom line is that each question was treated separately during analysis.

Did you group all the demographic data for survey 1 and 2? 

Figure 1 legend clearly states that the pie diagram was for survey 1 (and that survey 2 was not qualitatively different. Figure 1 clearly shows that Survey 1 and 2 were summarized separately for geographic region, gender and age.

No meaningful statistical analysis to indicate correlation between languages/ knowledge. 

We did Fisher exacts in the initial manuscript, but a reviewer pointed out to us that Fisher tests are more appropriate for small data sets and that chi square tests would be more appropriate for our dataset. Hence we recalculated all statistics with the chi-square test. The results of these tests for language are imbedded in Supplemental File 3.

No information on whether the validity and reliability of the questionnaires has been assessed either! 

We asked questions very similar to previous surveys and used those as a guideline to our questions. This was especially true of the questions about antibiotics. As we state in the manuscript :

Polling question design was guided by preliminary verbal surveys and prior observations of user kiosk experiences at the AMNH. 

However we do not present a formal analysis of those preliminary surveys. The preliminary surveys gave us information on how many questions we could ask without “irritating” the museum goer (answer is 7 to 10) and what level we needed to direct our questions.

Did the authors consider a specific sample size (calculating a sample size) for their survey?

Sample sizes are given on the SEPA-kiosk website which we reference several times in the manuscript. Basically, for considering a geographic region (this is the only category where sample sizes might be small as gender and age categories were quite large) we required an N=95 individuals. We did not attempt to calculate effective sample size with the dataset. 

Results:

You mention using the Fisher Exact test in the results section- such information should be included in the methods section as part of the data analysis. What is the rationale for using a cut-off point of N>95?

Any statistical test sets a cutoff and we could just as easily have set the cutoff at 90% 0r 99%. However, in general, most researchers are comfortable with the 95% cutoff. 

Table 5: unfortunately, I do not understand what is meant by NZ Egypt, NZ India etc the explanation is unclear.

NZ Egypt refers to New Zealanders migrating from Egypt to NZ and NZ India refers to New Zealanders migrating to NZ from India. This is now indicated more clearly in the legend to Table 5.

Figure legends should not include an explanation of the result- but just an explanation of what the figures show- the explanation and analysis should come within the text not the figure legend.

We have edited the figure legends to not have results in them.

The discussion brings out a few interesting threads however the overall conclusion and recommendations are very generalised when this shouldn’t be the case because of the diversity of the participants as well as the limitations mentioned. Perhaps it would be more realistic to have a recommendation based on the museum’s educational programmes/ sections etc to address this identified lack of knowledge in the first instance.

This is similar to R2’s request that we make a statement about the policy changes or issues that the study might be relevant to. We have added the following to the conclusion :

From the perspective of a natural history museum or a science center knowing where to focus efforts is a first step in developing functional and effective programs and exhibitions.

The current study was part of a larger initiative started at the AMNH in 2015 to educate the public about microbes and issues related to microbes that included two exhibitions on the microbiome and Science Café programming events as well. The surveys were designed by the museum’s educators, exhibition staff and scientific advisory panel to probe the museum goer’s knowledge of certain issues connected to microbial diversity and microbial life. The data from the surveys can inform the museum educators and exhibition staff as to what level materials developed by the outreach and informal education arms of the museum should be targeted at. The AMNH uses such information to create a system for gauging audience interest in particular topics, as well as identify gaps in audience knowledge between cutting-edge research on biodiversity and health and public perceptions of those intersections and their implications. The current results indicate that in developing future programming

We also suggest that the analysis here can be disseminated for the benefit of other formal and informal science institutions, providing them with methodology and data they can use for their own programming. This survey of public knowledge will be particularly valuable in influencing critical conversations on national science education and science policy, and dissemination efforts aimed to reach these relevant audiences. These results then can help guide the design of specific education programs. Public education programs about microbes often start with the somewhat sophisticated topic of antimicrobial resistance. We demonstrate a lack of general knowledge about microbes in the museum going public that suggest museum educators and perhaps even public health educators should reassess the level at which information about microbes is initially presented (Barberán et al., 2016; Cirstea et al., 2018). 

---

## [Decision Letter · Decision Letter 1]

6 Jul 2021

PONE-D-21-01340R1

A Natural History Museum Visitor Survey of Perception, Attitude and Knowledge (PAK) of Microbes and Antibiotics

PLOS ONE

Dear Dr. DeSalle,

Thank you for submitting your manuscript to PLOS ONE. After careful consideration, we feel that it has merit but does not fully meet PLOS ONE’s publication criteria as it currently stands. Therefore, we invite you to submit a revised version of the manuscript that addresses the points raised during the review process.

Your manuscript was assessed by one of the reviewers who originally assessed your manuscript. You will note that they have made two requests. However, compliance with these requests is not necessarily to meet PLOS ONE's publication criteria. *Updating your manuscript in response to the concerns noted by Reviewer 2 is therefore optional. *

However, throughout my own assessment of the manuscript I did note a number of revisions that will be required before publication:

A) In the previous round of review two reviewers noted that the following fragment was not supported by references "We point out that the results reported here are for museum goers, a category which is predominately better informed than others.". You have indicated in your response-to-reviewers that these references are provided in your manuscript. Whilst I understand that these are references are provided, this is some 6 paragraphs after the noted fragment. I would therefore would be grateful if you could provide appropriate references immediately following the statement "We point out that the results reported here are for museum goers, a category which is predominately better informed than others."

B) Your response to Reviewer 3 indicates that no formal analysis of the reliability/validity of the questions has been performed. Since this is an important part of any study using a questionnaire, please note the absence of these analyses in the limitations subsection of your Discussion.

We look forward to receiving your revised manuscript.

Kind regards,

George Vousden

Division Editor

PLOS ONE

Journal Requirements:

Reviewers' comments:

Reviewer's Responses to Questions

**Comments to the Author**

1. If the authors have adequately addressed your comments raised in a previous round of review and you feel that this manuscript is now acceptable for publication, you may indicate that here to bypass the “Comments to the Author” section, enter your conflict of interest statement in the “Confidential to Editor” section, and submit your "Accept" recommendation.

Reviewer #2: (No Response)

2. Is the manuscript technically sound, and do the data support the conclusions?

Reviewer #2: Yes

3. Has the statistical analysis been performed appropriately and rigorously? 

Reviewer #2: Yes

4. Have the authors made all data underlying the findings in their manuscript fully available?

Reviewer #2: Yes

5. Is the manuscript presented in an intelligible fashion and written in standard English?

Reviewer #2: Yes

6. Review Comments to the Author

Reviewer #2: 1. Since the authors said that museum goers don’t represent the lay public, why did they mention the public perception of antibiotics in the introduction? eg., “Obtaining basic information on how a lay audience perceives and reacts to these topics is necessary for understanding where to direct educational efforts”.

2. I still stand by my opinion. The first paragraph of the discussion should summarize the main findings of the study.

7. PLOS authors have the option to publish the peer review history of their article (what does this mean?). If published, this will include your full peer review and any attached files.

Reviewer #2: No

---

## [Author Response · Author response to Decision Letter 1]

27 Jul 2021

Response to reviewers 

PONE-D-21-01340R1

A Natural History Museum Visitor Survey of Perception, Attitude and Knowledge (PAK) of Microbes and Antibiotics

There were two points you wanted us to address and these are in yellow below. Our response to the two points are in blue. You will see that we complied entirely with your requests.

In the previous round of review two reviewers noted that the following fragment was not supported by references "We point out that the results reported here are for museum goers, a category which is predominately better informed than others.". You have indicated in your response-to-reviewers that these references are provided in your manuscript. Whilst I understand that these are references are provided, this is some 6 paragraphs after the noted fragment. I therefore would be grateful if you could provide appropriate references immediately following the statement "We point out that the results reported here are for museum goers, a category which is predominately better informed than others."

I have added the museum-goer references in the proper place and shifted the reference numbers accordingly.

B) Your response to Reviewer 3 indicates that no formal analysis of the reliability/validity of the questions has been performed. Since this is an important part of any study using a questionnaire, please note the absence of these analyses in the limitations subsection of your Discussion

Inserted this statement in “Limitations” section:

Any study using a survey is based on the reliability and validity of the questions. While no formal analysis of these survey parameters were accomplished we suggest that our pre survey exploration addressed some of the validity concerns that surveys face. We also suggest that the consistency of similar answers from the survey done in 2017 with those from 2018 indicate a degree of reliability of those survey questions. Finally, we point out that several of the questions in our survey have been posed before in other surveys on antibiotics and microbial issues, which also suggesting a degree of validity of this study.

Thanks for moving this manuscript along. We feel it is now in very good shape after the two rounds of reviews and subsequent revisions. We fund the review process very helpful and feel strongly that it makes the paper much better and clearer. If you have any other requests about this manuscript please contact me.

---

## [Editor Report · Decision Letter 2]

24 Aug 2021

A Natural History Museum Visitor Survey of Perception, Attitude and Knowledge (PAK) of Microbes and Antibiotics

PONE-D-21-01340R2

Dear Dr. DeSalle,

We’re pleased to inform you that your manuscript has been judged scientifically suitable for publication and will be formally accepted for publication once it meets all outstanding technical requirements.

Kind regards,

George Vousden

Division Editor

PLOS ONE
---

## [Editor Report · Acceptance letter]

13 Sep 2021

PONE-D-21-01340R2 

A Natural History Museum Visitor Survey of Perception, Attitude and Knowledge (PAK) of Microbes and Antibiotics 

Dear Dr. DeSalle:

I'm pleased to inform you that your manuscript has been deemed suitable for publication in PLOS ONE. Congratulations! Your manuscript is now with our production department. 

Kind regards, 

on behalf of

Dr. George Vousden 

Staff Editor

PLOS ONE